# The Differential Expression of Circular RNAs and the Role of circAFF1 in Lens Epithelial Cells of High-Myopic Cataract

**DOI:** 10.3390/jcm12030813

**Published:** 2023-01-19

**Authors:** Shiyu Ma, Xiangjia Zhu, Dan Li, Fan Yang, Jiaqi Meng, Yongxiang Jiang, Jing Ma, Yi Lu

**Affiliations:** 1Department of Ophthalmology, Eye and ENT Hospital, Shanghai Medical College, Fudan University, Shanghai 200031, China; 2Eye Institute, Eye and ENT Hospital, Shanghai Medical College, Fudan University, Shanghai 200031, China; 3Key Laboratory of Myopia, Ministry of Health PR China, Shanghai 200031, China; 4Shanghai Key Laboratory of Visual Impairment and Restoration, Shanghai Medical College, Fudan University, Shanghai 200031, China; 5Department of Facial Plastic and Reconstructive Surgery, ENT Institute, Eye and ENT Hospital, Shanghai Medical College, Fudan University, Shanghai 200031, China

**Keywords:** high-myopic cataract, circular RNA, lens epithelial cells, circAFF1, ceRNA

## Abstract

High-myopic cataract (HMC) is a complex cataract with earlier onset and more rapid progress than age-related cataract (ARC). Circular RNAs (circRNAs) have been implicated in many diseases. However, their involvement in HMC remain largely unexplored. To investigate the role of dysregulated circRNAs in HMC, lens epithelium samples from 24 HMC and 24 ARC patients were used for whole transcriptome sequencing. Compared with ARC, HMC had 3687 uniquely expressed circRNAs and 1163 significantly differentially expressed circRNAs (DEcRs) (|log_2_FC| > 1, *p* < 0.05). A putative circRNA-miRNA-mRNA network was constructed based on correlation analysis. We validated the differential expression of 3 DEcRs by quantitative polymerase chain reaction (qPCR) using different sets of samples. We further investigated the role of circAFF1 in cultured lens epithelial cells (LECs) and found that the overexpression of circAFF1 promoted cell proliferation, migration and inhibited apoptosis. We also showed that circAFF1 upregulated Tropomyosin 1 (TPM1) expression by sponging miR-760, which was consistent with the network prediction. Collectively, our study suggested the involvement of circRNAs in the pathogenesis of HMC and provide a resource for further study on this topic.

## 1. Introduction

High-myopic cataract (HMC) refers to a kind of cataract complicated by high myopia (HM) [1,2]. Compared to the most common age-related cataract (ARC), the typical characteristics of HMC included long eye axial length, high incidence of nuclear cataract, earlier onset and more rapid progression of the condition [3]. The population with HMC is becoming larger and younger with the increasing incidence of HM among adolescents [4]. Because HMC is associated with special pathological changes such as posterior scleral staphyloma and zonular weakness, the surgery for HMC is often more difficult than that for ARC, and the postoperative complications such as rupture of the zonules from lens-iris diaphragm retropulsion syndrome (LIDRS), retinal detachment and progressed myopic traction maculopathy are more likely to occur [5,6].

It has been generally accepted that the pathogenesis of HMC involves the interaction between genetic and environmental factors [7]. Variations in sequence or an abnormal expression of several genes have been reported in HMC. The homozygous mutations of Leprecan Like 1 (*LEPREL1*) were identified in families with HM and early-onset cataract [8,9]. The crystallin alpha A protein level was found decreased in the lens epithelium of HMC compared to ARC [10], which might be associated with HMC by weakening the interaction with unfolded proteins and aggravating protein aggregation [11]. Compared to ARC, HMC showed higher levels of transforming growth factor-β2 (TGF-β2) [12], interleukin-1 receptor antagonist (IL-1ra) [13], angiopoietin-1 (ANG-1) [14], matrix metalloproteinase (MMPs), tissue inhibitor of metalloproteinases (TIMPs) [15], and lower level of monocyte chemoattractant protein-1 (MCP-1) in the aqueous humor [13]. These differences suggested that HMC is also associated with changes in the microenvironment of the eye. However, the etiology and pathogenesis of HMC is still unclear.

Circular RNA (circRNA) is a type of non-coding RNA with a stable closed circular structure without 3′-poly(A) tail or 5′-cap [16]. As one of the hot research topics in epigenetics, circRNAs have been implicated in the development and progression of many ocular diseases such as diabetic retinopathy [17,18], neovascular age-related macular degeneration [19,20], glaucoma [21] and ARC [22,23]. However, the role of circRNAs in HMC has not been explored.

In the present study, we performed whole transcriptome sequencing on lens epithelium samples obtained from HMC and ARC patients and found that a large amount of circRNAs were differentially expressed in HMC. A competing endogenous RNA (ceRNA) network were constructed based on differentially expressed circRNAs (DEcRs), miRNAs (DEmRs) and mRNAs (DEGs). We further demonstrated that circAFF1 increased lens epithelial cells (LECs) viability and migration capacity, inhibited cell apoptosis, and stimulated the expression of Tropomyosin 1 (TPM1) by sponging miR-760. Collectively, our study suggested the involvement of circRNA in the development of HMC.

## 2. Materials and Methods

### 2.1. Patients and Samples

This study protocol was approved by the Ethics Committee of the Eye and ENT Hospital of Fudan University (approval number: 2014055) and adhered to the tenets of the Declaration of Helsinki. All patients underwent phacoemulsification surgery. We collected the would-be-discarded lens epithelium tissue from 54 HMC and 54 ARC patients after continuous curvilinear capsulorhexis in cataract surgery at our hospital during the period of November 2020 to September 2021. The type and the severity of cataract were classified according to the Lens Opacities Classification System III (LOCS III) [24]. ARC was defined as typical cortical cataract in eyes with axial length between 21.0–24.5 mm (LOCSIII C 3–5), whereas HMC was defined as typical nuclear cataract with axial length longer than 26 mm (LOCSIII NC 3–5). Patients with other eye conditions, current or previous ocular trauma and positive ocular surgery history were excluded. The basic information of patients involved in the study were shown in Table 1.

Samples were kept in RNA stabilization reagent (Solarbio, Beijing, China) at −80 °C until further use. Twenty-four samples from HMC and ARC each were used for whole transcriptome sequencing, and the rest 30 samples from each group were used for quantitative polymerase chain reaction (qPCR) analysis.

### 2.2. RNA Extraction and Whole Transcriptome Sequencing

In order to obtain enough RNA for whole transcriptome sequencing, eight samples in each group were mixed and counted as one biological replicate. RNA was extracted using TRIzol reagent (Invitrogen, CA, USA). The quality and quantity of total RNA was determined by Agilent 2100 Bioanalyzer (Agilent Technologies, Palo Alto, CA, USA) and NanoDrop (Thermo Fisher Scientific, Wilmington, DE, USA). One microgram of ribosomal-depleted RNA with RNA integrity number (RIN) value above 7 was obtained using Ribo-Zero rRNA removal Kit (Illumina, San Diego, CA, USA) for library construction with NEBNext^®^ Ultra™ Directional RNA Library Prep Kit (Illumina). For small RNAs, 2 µg of total RNA with RIN value above 7.5 was used for library preparation using NEBNext^®^ Multiplex Small RNA library Prep Set (Illumina). Libraries with different indices were multiplexed and loaded on HiSeq instrument (Illumina). The whole transcriptome sequencing was conducted by Medical Laboratory of Nantong Zhongke (Nantong, China).

### 2.3. Sequencing Data Analysis

Raw data of fastq format were processed by Trimmomatic (v0.30, Bolger A M, MPlof Molecular Plant Physiology, Berlin, Germany) to filter redundant and low-quality data. For mRNAs, sequences were aligned to reference genome via hisat2 (v2.0.1, Daehwan Kim, Dallas, TX, USA), and gene and isoform expression levels were estimated by Stringtie (v1.3.5, Sam Kovaka, Baltimore, USA). CircRNAs were identified by CIRI (v2.0, Yuan Gao, Beijing, China) and their expressions were calculated according to the junction reads at the back-splicing loci. Spliced reads per billion mapping (SRPBM) was used to normalize the circRNA reads. MiRDeep2 (v0.0.8, Sebastian Mackowiak, Berlin, Germany) was used to identify miRNA and evaluate their expressions. Differential expression analysis was performed with the DEGSeq Bioconductor package (v1.42.0, Likun Wang, Beijing, China). After using Storey’s way for controlling the false discovery rate, corrected *p*-values were denoted as *q*-values. Criteria for differential expression was set uniformly for mRNAs, miRNAs and circRNAs as follows: *q*-value < 0.05 and |log_2_fold change (FC)| > 1.

Hierarchical clustering analysis for DEcRs and DEGs was performed using Cluster (v2.1.1, Michiel J. L. De Hoon, Tokyo, Japan). Gene Ontology (GO) and Kyoto Encyclopedia of Genes and Genomes (KEGG) analysis was performed using clusterProfiler (v3.16.1, Guangchuang Yu, Southern Medical University, Guangzhou City, China) with the following parameters: minGSSize = 10, maxGSSize = 500, *p*-value < 0.05.

### 2.4. CeRNA Network Construction

CeRNA regulatory network among DEcRs, DEmRs and DEGs was constructed according to the following procedures: Firstly, target mRNAs and circRNAs of miRNA were identified using miRanda (v3.3a, ChenLiang, Macau S.A.R., China) with thresholds: SCORE ≥ 150 and ENERGY ≤ −20. Pearson correlation coefficient (PCC) of miRNA-circRNA and miRNA-mRNA interactions were calculated. The miRNA-circRNA and miRNA-mRNA pairs with PCC < −0.8 and *p*-value < 0.05 were selected, and the co-expressed mRNAs and circRNAs that were regulated by the same miRNAs were identified with the criteria of PCC > 0.8 and *p*-value < 0.05. Finally, Cytoscape (v3.6.1, Otasek, La Jolla, CA, USA) software was utilized to visualize the ceRNA network.

### 2.5. qPCR Assay

Total RNA was extracted from lens epithelium samples or LECs with TRIzol reagent (Invitrogen) and reverse transcribed to cDNA using the Prime Script RT Reagent Kit with gDNA Eraser (Yisheng Biotech, Shanghai, China). qPCR was performed with SYBR Premix Ex Taq™ (Takara, Shiga, Japan) on a StepOnePlus™ Real-Time PCR System (Thermo Fisher Scientific). Glyceraldehyde 3-phosphate dehydrogenase (GAPDH) was used as internal control. The relative expression levels of validated circRNAs were calculated by 2^−ΔΔCt^ method. The sequence of all primers used were listed in Table 2.

### 2.6. Cell Culture

Human embryonic kidney 293T (HEK293T) cells and human LEC line (SRA01/04) were cultured in Dulbecco’s modified Eagle’s medium with 10% fetal bovine serum (Biological Industries, Kibbutz, Beit Haemek, Israel) and 1% penicillin/streptomycin (NCM Biotech, Suzhou, China) at 37 °C with 5% CO_2_.

### 2.7. Plasmid Construction and Dual-Luciferase Reporter Assay

The following plasmids were constructed for the transfection in HEK293T cells: psiCHECK containing wildtype circAFF1 (predicted binding sequence fragment: 5′-gcCCCAGGAGCACAGAGCCc-3′, 293T^WTcAFF1^) and mutated circAFF1 (mutant sequence: 5′-gcTAAGGGAGCATGCGTATc-3′, 293T^MUTcAFF1^). The plasmids were then used to transfect HEK293T cells, with the co-transfection of miR-760 mimics or control mimics using Lipofectamine 2000 (Invitrogen). Cells were lysed 48 h (hrs) later. Both firefly and Renilla luciferase activities were measured using a dual-luciferase reporter assay system (Promega, Madison, WI, USA), and the Renilla luciferase activities were normalized to firefly luciferase activities.

### 2.8. RNA Fluorescence In Situ Hybridization (FISH) Assay

LECs were seeded in a 4-chanmber slide for 24 h, fixed with 4% paraformaldehyde (PFA), permeabilized with TritonX-100 and incubated with RNA probes at 42 °C in the dark overnight. The slides were washed with 2× saline sodium citrate (SSC) buffer, then the nuclei were counterstained with 4′-6-diamidino-2-phenylindole (DAPI, Cell Signaling Technology, Boston, MA, USA). An EVOSTM Microscope M5000 Imaging System (Invitrogen) was used for visualization. The U6 and 18S ribosome probes were used as nuclei and cytoplasm controls, respectively. The circAFF1 probe and control probe were designed and synthesized by RiboBio Co., Ltd. (Guangzhou, China).

### 2.9. Construction of Stably Transfected Cell Lines by Lentivirus Transfection

The plasmids carrying circAFF1 (pLC5-ciR-circAFF1) and control (pLC5-ciR) were generated by Geneseed (Guangzhou, China) and the sequences were validated. These plasmids were used to transfect HEK293T cells with L-PEI (Yisheng Biotech) for 48 h to obtain lentivirus. The lentivirus was purified and used to infect LECs. The stably transfected cells were obtained after puromycin screening for 3 days and then used for subsequent analysis as indicated.

### 2.10. Cell Proliferation Assay

Cell proliferation was analyzed by Cell Counting Kit 8 (NCM Biotech) according to the manufacturer’s instructions. Cells were seeded in 96-well plates at a final density of 5 × 10^3^/well. Ten microliters of CCK-8 solution was added to each well 2 h before the termination of incubation. The optical density (OD) values at 450 nm were measured by a microplate reader (Biotek, Winooski, VT, USA) at 0, 24, 48, 72 and 96 h. The cell proliferation rate was calculated relative to the value of OD_450nm_ at 0 h.

### 2.11. 5-Ethynyl-2′-deoxyuridine (EdU) Assay

The EdU assay was used to quantitate proliferating cells following the manufacturer’s instruction (RiboBio Co.). Cells in 96-well plates were incubated with 50 µM EdU for 2 h and fixed with 4% PFA. After Apollo and Hoechst staining, cells were visualized and photographed by EVOSTM Microscope M5000 Imaging System.

### 2.12. Transwell Cell Migration Assay

Cells were seeded in the upper chamber of Transwell inserts with 200 µL medium without FBS. One milliliter medium with 20% FBS was added to the lowed chamber. After 48 h at 37 °C, cells were fixed with 4% PFA and stained with 2% crystal violet staining solution. The migrated cells were captured under microscope and counted by ImageJ.

### 2.13. Wound Healing Assay

Cells in 6-well plate were incubated with 5% CO_2_ at 37 °C until near 100% confluency. A cell scratcher was used to create a wound in the middle of the well. Images were captured immediately after the wounding under microscope (0 h) and the width of the wound was measured. They were then cultured with serum-free medium for 36 h and photographed again. The relative migration rate was calculated as follows: (the average width of the wound at 0 h—the width at 36 h)/the width at 0 h × 100%.

### 2.14. Cell Apoptosis Assay

Cell apoptosis was analyzed using the Annexin V-FITC/propidium iodide (PI) detection kit (Signalway Antibody, College Park, MD, America) according to manufacturer’s instruction and quantitated using flow cytometer (Beckman BD Celesta, San Francisco, CA, USA).

### 2.15. Statistical Analysis

All experiments were repeated at least three times. Statistical analysis was performed using un-paired two-tailed Student’s *t*-tests via GraphPad Software (v9.3.1, La Jolla, CA, USA). Data were presented as mean ± standard deviation (SD). Statistical significance was accepted at *p*-value < 0.05.

## 3. Results

### 3.1. DEcRs in Lens Epithelium between HMC and ARC

A total of 9192 circRNA were detected by whole transcriptome sequencing across all lens epithelium samples, consisting of 7091 circRNAs in the HMC group and 5505 circRNAs in the ARC group (Figure 1A,B). Among them, 84.0% of circRNAs were derived from the exons, 8.8% were intronic and 7.2% were intergenic circRNAs (Figure 1C). The length of most circRNAs was less than 800 nucleotides (NTs) (Figure 1D) and they were mainly distributed on chromosomes 1, 2 and 3 (Figure 1E).

We found 2101 (22.9%) circRNAs uniquely expressed in ARC and 3687 (40.1%) circRNAs uniquely expressed in HMC. Within 3404 (37.0%) circRNAs that were expressed by both samples, 1163 were significantly differentially expressed between these two groups (510 upregulated and 653 downregulated). The heatmap showed the top 20 upregulated and downregulated circRNAs in HMC (Figure 1F). The data of whole transcriptome sequencing has been uploaded to Gene Expression Omnibus (GEO accession number PRJNA785074).

### 3.2. Potential Functions of DEcRs and DEGs in HMC

GO analysis showed that the DEcRs of HMC are mainly enriched in biological processes (BP) related to chromatin and protein modification, such as covalent chromatin modification, histone modification and protein polyubiquitination (Figure 2A), whereas DEGs of HMC were enriched in protein targeting and RNA catabolic process (Figure 2B). KEGG analysis showed that the enriched pathways of DEcRs (Figure 2C) and DEGs (Figure 2D) of HMC included the TGF-β [25], Wnt [26] and mTOR [27] signaling pathways, which have been reported to play important roles in the initiation and progression of cataract.

Based on the correlation analysis among DEcRs, DEmRs and DEGs of HMC, we constructed a ceRNA network including 120 circRNAs, 47 miRNAs and 102 mRNAs (Figure 2E). There were 289 ceRNA pairs in the network. Some DEcRs were predicted to interact with multiple miRNAs and corresponding mRNAs. For example, hsa_circ_0070386 (circAFF1), hsa_circ_0073171 (circMSH3), hsa_circ_0048747 (circSAFB2), hsa_circ_0089972 (circREPS2) and hsa_circ_0000798 (circBPTF) were predicted to interact with miR-760, and miR-760 may interact with 7 mRNAs including Chromosome 7 Open Reading Frame 50 (C7orf50), Golgi SNAP Receptor Complex Member 2 (GOSR2), KxDL Motif Containing 1 (KXD1), Metastasis Associated 1 Family Member 3 (MTA3), Pleckstrin And Sec7 Domain Containing 3 (PSD3), Tensin 2 (TNS2) and TPM1.

### 3.3. Validation of Differential circRNAs Expression in HMC by qPCR

To validate the sequencing data above, we picked four circRNAs for further qPCR analysis: circSLC15A4, which was expressed only in HMC; circDENND5B, which was expressed only in ARC; circAFF1, which was expressed in both HMC and ARC but was significantly upregulated in the former; circTRPM7, which was also expressed in both HMC and ARC but was significantly downregulated in the former. qPCR analysis was performed using a separate set of lens epithelium samples as described in the Methods (Table 3). The results showed a 3.34 ± 0.97 fold increase (*p* = 0.003) of circAFF1 expression and a 2.39 ± 0.87 fold increase (*p* = 0.016) of circSLC15A4 expression in HMC group compared to ARC. The expression of circDENND5B showed a 80.33 ± 9.27% decrease (*p* = 0.003) in HMC compared with ARC (Figure 3).

### 3.4. Overexpression of circAFF1 Promoted Cell Proliferation and Migration and Inhibited Cell Apoptosis in LECs

To further illustrate the role of circRNA, we chose circAFF1 to study its function in LECs. RNA-FISH revealed that circAFF1 was distributed in the cytoplasm of LECs (Figure 4A). We generated stable transfected LECs with overexpression of circAFF1 (LEC^cAFF1^) and used vector (LEC^vector^) as control. qPCR analysis showed 31.89 ± 2.55 (mean ± SD) times increased expression of circAFF1 in LEC^cAFF1^ compared to LEC^vector^ (*n* = 3, *p* < 0.0001, unpaired *t*-test, Figure 4B). Next, we compared proliferation of LEC^cAFF1^ and LEC^vector^. LEC^cAFF1^ showed a small yet significant higher proliferation rate than controls starting at 24 h (18.22 ± 4.69% higher than control, average ± SD, *n* = 3, *p* = 0.005, unpaired *t*-test) and lasted until the end of the experiment, which as 96 h later (29.68 ± 1.44% higher than control, average ± SD, *n* = 3, *p* < 0.0001, unpaired *t*-test, Figure 4C). Consistently, EdU incorporation analysis showed that the percentage of proliferating LEC^cAFF1^ cells (42.44 ± 1.59%) was 1.41 ± 0.12 times of the LEC^vector^ (30.24 ± 1.81%) (*n* = 3, *p* = 0.002, unpaired *t*-test, Figure 4D).

Next, we analyzed the motility of LEC^cAFF1^. The Transwell assay showed 1.35 ± 0.09 times increase of migrated LEC^cAFF1^ cells (374.67 ± 31.35) compared to LEC^vector^ (277.33 ± 8.73) (*n* = 3, *p* = 0.013, unpaired *t*-test, Figure 4E). Consistently, wound healing assay also showed that LEC^cAFF1^ cell migration was increased to 247.87 ± 7.8% of the control cells (*n* = 3, *p* = 0.0076, unpaired *t*-test, Figure 4F).

Finally, we compared the number of apoptotic cells by flow cytometry analysis. In LEC blank control, we counted 4.82 ± 1.86% apoptotic cells. On the other hand, apoptotic cells were 12.11 ± 0.26% in LEC^vector^, and 7.22 ± 0.43% in LEC^cAFF1^. Therefore, LEC^cAFF1^ showed 40.44 ± 2.3% decreased cell apoptosis compared to LEC^vector^ (*n* = 3, *p* < 0.001, unpaired *t*-test, Figure 4G).

### 3.5. CircAFF1 Regulate *TPM1* Expression by Sponging miR-760

The ceRNA network presented above predicted that circAFF1 may interact with miR-760 through the potential binding sites to regulate TPM1 expression. To validate the prediction, we transfected HEK293T cells with a plasmid containing circAFF1 and miR-760 mimics and examined the luciferase activity. Compared to cells with the control miRNA mimics, we observed 65.43 ± 4.79% decrease in luciferase activity (*n* = 3, *p* = 0.003, unpaired *t*-test). However, when circAFF1 with mutated binding sequence was introduced to HEK293T cells, we observed no significant change in luciferase activity (Figure 5A). The results suggested the interaction between circAFF1 and miR-760 via the predicted binding sequences.

Next, we examined the expression of miR-760 and TPM1 in LECs by qPCR. The expression of miR-760 decreased 35.21 ± 5.07% while the expression of TPM1 increased 198.28 ± 27.96% when circAFF1 was successfully overexpressed in LECs. Meanwhile, the overexpression of miR-760 in LEC^cAFF1^ downregulated the expression of TPM1 by 81.86 ± 1.19% (Figure 5B). The results suggested that circAFF1 regulated TPM1 expression through the inhibition of miR-760.

## 4. Discussion

More and more evidence suggested the involvement of circRNAs in the occurrence and development of cataract [28]. For example, CircMRE11A was involved in ARC by interacting with UBXN1 to promote cell cycle arrest and aging of LECs [29]. CircZNF292 [30], circHIPK3 [23], circEBP41 [31], circ0122396 [32], circKMT2E [33] and circPAG1 [34,35] regulate cell proliferation, apoptosis, autophagy or oxidative stress response by targeting different miRNAs, and thus promote the progression of ARC and diabetic cataract (DC). Moreover, the reduced m6A abundance of circRNAs in the lens epithelium of ARC patients could affect the stability, localization, cleavage, and translation of circRNAs [36]. However, the HMC related circRNA expression profile had not been reported.

In this study, we obtained circRNA profile of HMC and found that HMC was accompanied by significant changes in circRNA expression when compared to ARC. These circRNAs were enriched in biological processes which controls chromatin and protein modification and enriched in many cataract-related KEGG pathways. A network of circRNA-miRNA-mRNA was predicted. Our results provided a useful resource for future studies on the roles of circRNA in the pathogenesis of HMC. The different expression of circRNAs may contribute to the pathogenic processes of HMC depending on their localization and interactions with DNA, RNA and proteins [37]. For example, circRNA may regulate the HMC related gene expression such as *CRYAA* [10] by modulating transcription [38], sponging miRNA [30] or interact with mRNA directly [39]. Besides, circRNA could also be the sponges for proteins [40]. CircRNAs may bind proteins and act as scaffolding molecules that regulate protein-protein interactions [41], which may influence the function of proteins such as α crystallin [10].

We further demonstrated the role of circAFF1 in regulating the activity of LECs. The excessive proliferation and migration of LECs could potentially lead to the aggregation and differentiation of LECs at the lens equator, which would accelerate the compaction and hardening of the lens nucleus [42]. We found overexpression of circAFF1 in LECs cell line SRA01/04 significantly promote the proliferation and migration and inhibit cell apoptosis, which suggested that the upregulation of circAFF1 in HMC lens capsule may accelerate the occurrence and development of HMC.

We further provided evidence to show that circAFF1 interacts with miR-760 and regulates the expression of TPM1. We found that circAFF1 was mainly distributed in cytoplasm of LECs, where most circRNAs function as miRNA sponges. The miR-760 was predicted to bind to circAFF1, which was downregulated in HMC lens epithelium samples and circAFF1 overexpressed LECs. The interaction was testified by dual luciferase experiments. TPM1 was the downstream gene of miR-760 [43], which was upregulated in lens epithelium samples of HMC according to the analysis of whole transcriptome sequencing. Previous studies found that TPM1 was upregulated in LECs isolated from cataractous lenses of Shumiya Cataract Rats (SCRs), compared with non-cataractous lenses [44]. TPM1 is a member of the tropomyosin family involved in regulating and stabilizing cell cytoskeleton actin filaments [45]. It was found to be induced by TGFβ-2 during epithelial-mesenchymal transition (EMT) in LECs, which has been proposed as a major cause of posterior capsule opacification (PCO) [46]. But the role of TPM1 in the pathogenesis of HMC needs to be further studied.

## 5. Conclusions

In conclusion, our study mapped the circRNAs in lens epithelium samples of HMC, and provided evidence to show that circAFF1 upregulated the expression TPM1 via miRNA-760 and participated in the regulation of LECs proliferation, apoptosis and migration. Further investigations on HMC-specific circRNAs may provide useful insights on the pathogenesis of HMC.

## Figures and Tables

**Figure 1 jcm-12-00813-f001:**
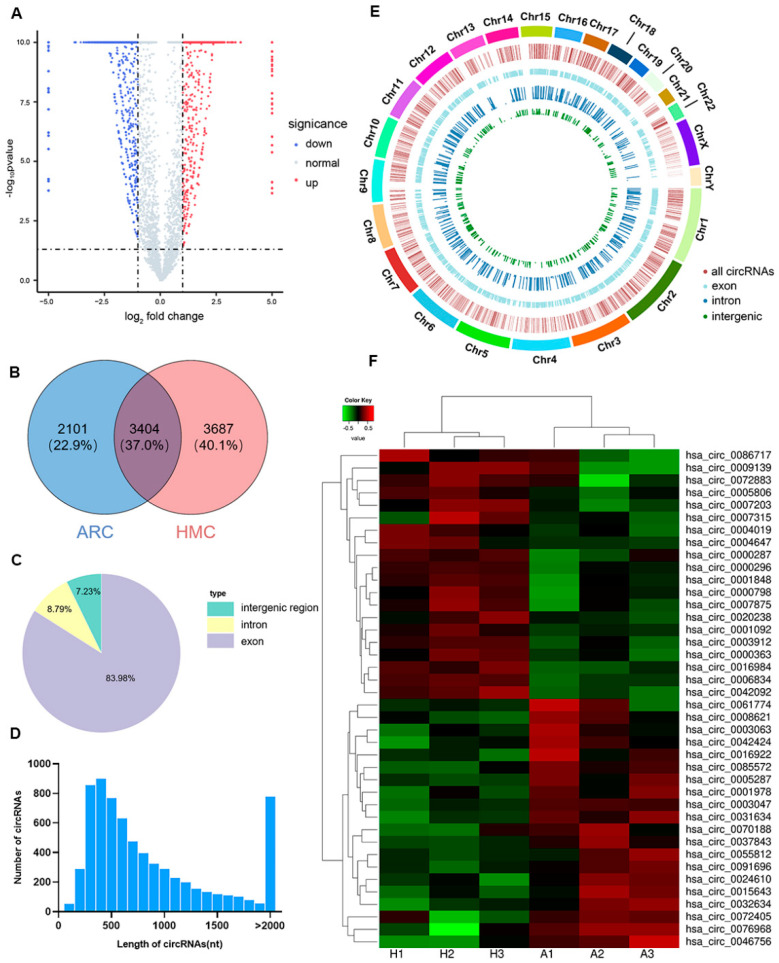
Total circRNAs detected by whole transcriptome sequencing in HMC and ARC lens epithelium samples. (**A**) Volcano plot of circRNAs. (**B**) The Venn diagram of overlapped circRNAs between HMC and ARC. (**C**) The genomic location of circRNAs. (**D**) The length distribution of circRNAs. (**E**) The chromosome distribution of circRNAs. (**F**) Hierarchical clustering analysis of the top 20 upregulated and downregulated DEcRs in HMC compared to ARC. Expression values are indicated by red and green shades, suggesting expressions above and below median expression level across all samples, respectively. H: HMC; A: ARC.

**Figure 2 jcm-12-00813-f002:**
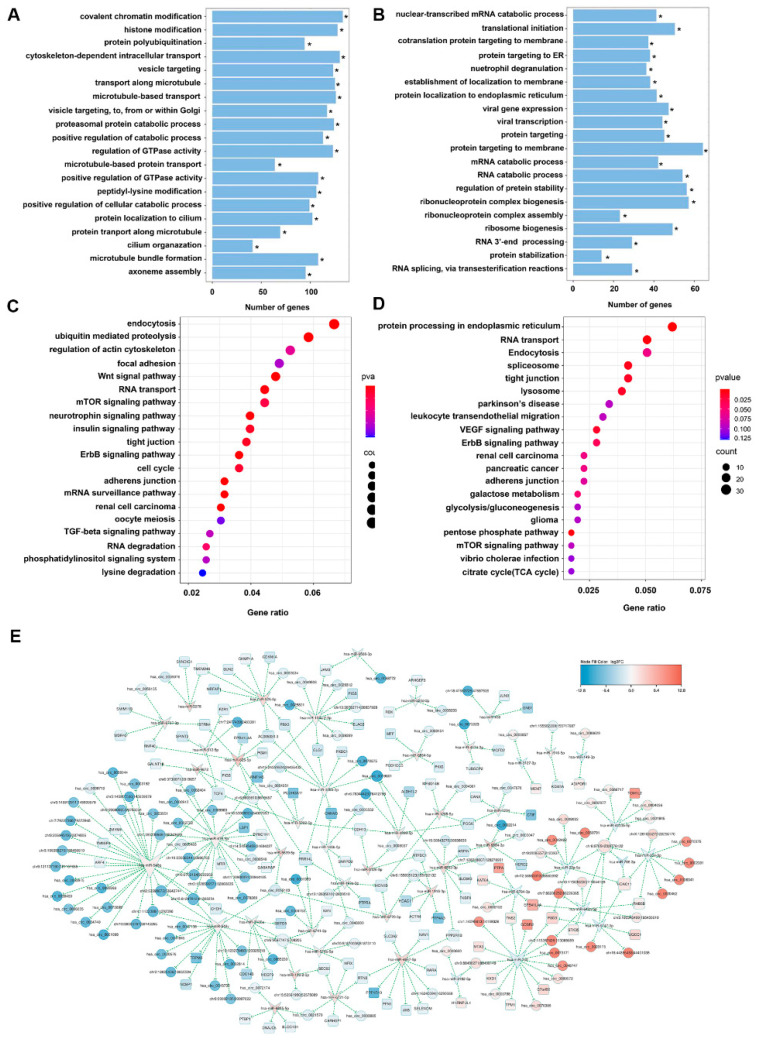
Functional enrichment analysis of DEcRs and DEGs and the predicted ceRNA network in HMC. (**A**) Enriched biological processes of GO by DEcRs. * *p* < 0.05. (**B**) Enriched biological processes of GO by DEGs. * *p* < 0.05. (**C**) Enriched KEGG pathways by DEcRs. (**D**) Enriched KEGG pathways by DEGs. (**E**) The ceRNA network. The association between miRNA, miRNA and mRNA was predicted based on correlation analysis according to the following standards: |log_2_FC| > 1 and *p* < 0.05, |PCC| > 0.8. Red node: upregulated circRNA, blue node: downregulated circRNA, blue arrows: downregulated miRNA, red arrows: upregulated miRNA, blue squares: downregulated mRNA, red squares: upregulated mRNA.

**Figure 3 jcm-12-00813-f003:**
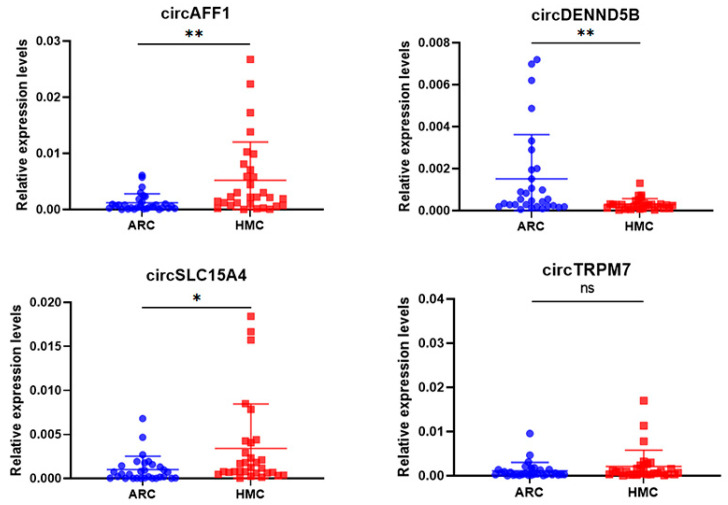
The expression of four selected circRNAs in HMC and ARC lens epithelium samples. RNA was extracted from 30 HMC and 30 ARC lens epithelium samples and used for qPCR analysis of the indicated circRNAs. The expression was normalized to GAPDH. Data were expressed as mean ± SD. ns *p* > 0.05, * *p* < 0.05, ** *p* < 0.01.

**Figure 4 jcm-12-00813-f004:**
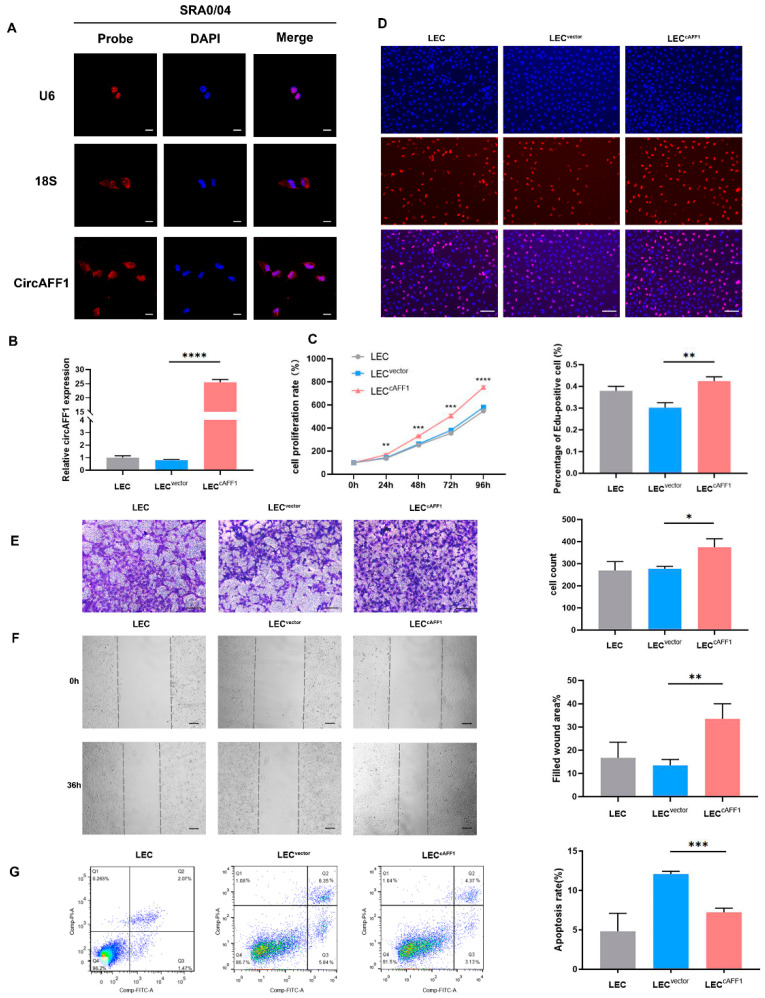
Overexpression of circAFF1 promoted cell proliferation, migration and inhibited cell apoptosis in LECs. (**A**) Subcellular localization of circAFF1 in LECs as determined by RNA-FISH assay. The nucleus was stained with DAPI. Scale bar: 20 μm. (**B**) Levels of circAFF1 expression in LECs with stable transfection. (**C**) The results of CCK-8 assay showing higher proliferation rate of LEC^cAFF1^ than LEC^vector^ after 24 h. *n* = 3, ** *p* < 0.01, *** *p* < 0.001, **** *p* < 0.0001. (**D**) The results of EdU incorporation showing more proliferating LEC^cAFF1^ than LEC^vector^. The quantitation of the labeled cells was shown in the panel below. *n* = 3, ** *p* < 0.01. Scale bar: 100 μm. (**E**) Transwell assay showed increased migration of LEC^cAFF1^ compared to LEC^vector^. *n* = 3, * *p* < 0.05. Scale bar: 100 μm. (**F**) Wound healing assay showed that the LEC^cAFF1^ migrated significantly faster than the LEC^vector^. *n* = 3, ** *p* < 0.01. Scale bar: 100 μm. (**G**) Cell apoptosis assay showed less apoptotic LEC^cAFF1^ than LEC^vector^. *n* = 3, *** *p* < 0.001. Data were expressed as mean ± SD.

**Figure 5 jcm-12-00813-f005:**
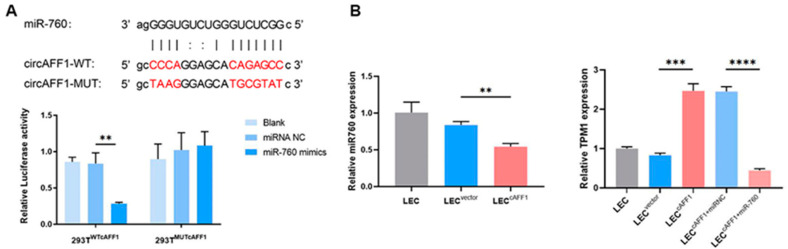
CircAFF1 increased TPM1 expression by sponging miR-760. (**A**) The upper panel: the predicted binding sequences of circAFF1 and miR-760; the lower panel: the relative luciferase reporter activity of 293T^WTcAFF1^ and 293T^MUTcAFF1^ following transfection with miR-760 mimics and control mimics. ** *p* < 0.01. (**B**) qPCR showed decreased expression of miR-760 in LEC^cAFF1^. The expression of TPM1 was increased in LEC^cAFF1^ and decreased in LEC^cAFF1^ with miR-760 mimics. ** *p* < 0.01, *** *p* < 0.001, **** *p* < 0.0001.

**Table 1 jcm-12-00813-t001:** Basic information of lens epithelium sample donors involved in this study.

	HMC	ARC
Number	54	54
Age at surgery (years) **	63.04 ± 7.97	69.29 ± 3.62
Gender (M/F)	17/37	19/35
Axial length (mm) ***	29.31 ± 2.40	23.28 ± 0.65
LOCS III grade	NC 3.71 ± 0.79	C 3.58 ± 0.76

HMC: high-myopic cataract; ARC: age-related cataract; M: male; F: female; NC: nuclear color; C: cortical cataract; Data were presented as mean ± SD where applicable; ** *p* < 0.01, *** *p* < 0.001.

**Table 2 jcm-12-00813-t002:** Primers used for qPCR analysis.

	Forward Primer (5′→3′)	Reverse Primer (5′→3′)
circAFF1	GCCAGAGGAAAGCAGAAGATAA	GTCCAGCTGCCATTTGTTTG
circDENND5B	GGAACAGATGCAGAACTTTGAC	GGATAGTGGGCGAGAACTTT
circSLC15A4	GTATTCCTGCTGTTCCCAGAA	CTCCCAGGTTAATGCTCCAATA
circTRPM7	TTGGGTCAGATGAACATCAAGATA	ACACATTCCCTCTTGGTCAAA
GAPDH	GGAGCGAGATCCCTCCAAAAT	GGCTGTTGTCATACTTCTCATGG

**Table 3 jcm-12-00813-t003:** The information of selected circRNAs.

circRNA	circBaseID	Location	Length (bp)	Trend	log_2_FC	Predicted Target miRNA	Trend	log_2_FC
circAFF1	hsa_circ_0070386	chr4:88035519|88036451	933	up	3.2	hsa-miR-760	down	−2
circDENND5B	hsa_circ_0025830	chr12:31604874|31648853	1502	down	−11.51	hsa-miR-103a-2-5p	up	1.26
circSLC15A4	hsa_circ_0000462	chr12:129299320|129299615	296	up	12.44	hsa-miR-204-3p	down	−1.01
circTRPM7	hsa_circ_0035249	chr15:50923614|50955243	1201	down	−2.3	hsa-miR-520d-3p	up	1.13

## Data Availability

The data of whole transcriptome sequencing presented in this study are openly available in Gene Expression Omnibus. GEO accession number: PRJNA785074.

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
