# Peer review of "The Differential Expression of Circular RNAs and the Role of circAFF1 in Lens Epithelial Cells of High-Myopic Cataract"

_jcm, 2023, doi:10.3390/jcm12030813_

Round 1
Reviewer 1 Report
In the present manuscript, the authors performed the whole transcriptome sequencing on lens epithelium samples from both HMC and ARC patients and found that a high amount of circRNAs was differentially expressed in HMC. In addition, they investigated the role of a circRNA shared by both HMC and ARC but significantly up-regulated in HMC, namely, cirAFF1, in lens epithelial cells of HMC. I strongly believe that the results from this study will provide a large contribution to the understanding of HMC. Therefore, I recommend publication after the following points have been addressed:
1. In section 3.3 of the results part, the authors chose circALC15A4 as one uniquely expressed cirRNA in HMC. However, later in the same paragraph the authors referred to this cirRNA as circSCL15A4 and on Table 3, it is named as circSLC15A4. Why these discrepancies in the codes? I believe there are typos issues here. Please fix them.
In addition, a cirRNA that is expressed in both HMC and ARC but significantly up-regulated in ARC must be included to have the full picture.
2. The figure in section 3.4 must be renumbered as Fig. 4 (currently Fig. 5) and the corresponding references in the text (Fig. 5A, 5B, ...) must be changed to Fig. 4A, 4B, ....
3. The figure in section 3.5 is numbered correctly but the corresponding references in the text must be changed from Fig. 6A, 6B to Fig. 5A, 5B.
Author Response
In the present manuscript, the authors performed the whole transcriptome sequencing on lens epithelium samples from both HMC and ARC patients and found that a high amount of circRNAs was differentially expressed in HMC. In addition, they investigated the role of a circRNA shared by both HMC and ARC but significantly up-regulated in HMC, namely, cirAFF1, in lens epithelial cells of HMC. I strongly believe that the results from this study will provide a large contribution to the understanding of HMC. Therefore, I recommend publication after the following points have been addressed:
Point 1:In section 3.3 of the results part, the authors chose circALC15A4 as one uniquely expressed cirRNA in HMC. However, later in the same paragraph the authors referred to this cirRNA as circSCL15A4 and on Table 3, it is named as circSLC15A4. Why these discrepancies in the codes? I believe there are typos issues here. Please fix them.
In addition, a cirRNA that is expressed in both HMC and ARC but significantly up-regulated in ARC must be included to have the full picture.
Response 1:Thank you very much for your correction. It shoud be circSLC14A4. We apologize for our negligence. We have corrected the wrong spellings in the revised manuscript. In addition, we verified the expression of circTRPM7 by qPCR, which was expressed both in HMC and ARC but significantly down-regulated in HMC. However, there was no difference in circTRPM7 expression between ARC and HMC. Such discrepancy was often seen between bulk-transcriptomic data and individual qPCR analysis. We added the result of circTPRM7 in the revised manuscript.
Point 2:The figure in section 3.4 must be renumbered as Fig. 4 (currently Fig. 5) and the corresponding references in the text (Fig. 5A, 5B, ...) must be changed to Fig. 4A, 4B, ....
Response 2:Thank you very much for pointing out the mislabeling. We have matched the figures with legends in the text in the revised manuscript. The wrong pictures and sentences are corrected now.
Point 3:The figure in section 3.5 is numbered correctly but the corresponding references in the text must be changed from Fig. 6A, 6B to Fig. 5A, 5B.
Response 3:Thank you very much for your correction. The mislabeled figures and texts are corrected now.

Reviewer 2 Report
The manuscript is peppered with expressions that might lend themselves well to possible revision
Samples were kept in RNA later
Tenets rather than tenet
Rupture of the suspensory ligament - Zonules from LIDRS?
Corneal endothelial decompensation not necessarily a frequent direct complication in high myopes unless surgeries become complicated.
Can the authors kindly elaborate on why the distinction between cortical cataracts in ARC and nuclear cataracts in HMC? Often the cataracts are not one or the other but a mix of cataract sub-types.
How were lens epithelium samples retrieved? Did all patients undergo phacoemulsification surgery?
I would have liked for the authors to discuss in greater detail the role of circRNA profiles of HMC identified and how these contribute to current pathogenic processes. This would be the crux of the topic being discussed in this paper.
Author Response
Point 1:The manuscript is peppered with expressions that might lend themselves well to possible revision
Samples were kept in RNA later
Tenets rather than tenet
Rupture of the suspensory ligament - Zonules from LIDRS?
Response 1:Thank you very much for your corrections. We’ve corrected the above inappropriate expression in the revised manuscript.
Point 2:Corneal endothelial decompensation not necessarily a frequent direct complication in high myopes unless surgeries become complicated.
Response 2:Thank you very much for your suggestion. We deleted “Corneal endothelial decompensation” in the revised manuscript.
Point 3:Can the authors kindly elaborate on why the distinction between cortical cataracts in ARC and nuclear cataracts in HMC? Often the cataracts are not one or the other but a mix of cataract sub-types.
Response 3:Thank you very much for your suggestions. As you correctly pointed out, cataracts are usually a mix of cataract subtypes. On the other hand, there are studies which suggested that high myopia is largely related to nuclear cataract, especially when age and gender were controlled[1-4] . However,this does not exclude the cortical involvement. To better distinguish ARC and HMC, we included ARC and HMC by mainly cortical involvement and typical nuclear involvement, respectively, in addition to the differences in axial length.
We revised the manuscript accordingly in Materials and Methods under Patients and Samples.
Point 4:How were lens epithelium samples retrieved? Did all patients undergo phacoemulsification surgery?
Response 4:All patients involved in this study underwent phacoemulsification and the lens epithelium tissues were collected after continuous circular capsulorhexis. We added a more detailed description in the Materials and methods——Patients and samples.
Point 5:I would have liked for the authors to discuss in greater detail the role of circRNA profiles of HMC identified and how these contribute to current pathogenic processes. This would be the crux of the topic being discussed in this paper.
Response 5:Thank you very much for your suggestion. CircRNAs could contribute to the pathogenic processes of HMC depending on their localization and interactions with DNA, RNA and proteins[5]. For example,circRNA may regulate the HMC related gene expression such as CRYAA[6] by modulating its transcription[7],sponging miRNA[8] or interact with mRNA directly[9]. Besides, circRNA could also be the sponges for proteins[10]. CircRNAs may bind proteins and act as scaffolding molecules that regulate protein-protein interactions[11], which may influence the function of proteins which maintain the transparency of lens such as α crystallin[12].
We found that overexpression of circAFF1 significantly promoted the proliferation and migration of LECs, while inhibiting cell apoptosis simultaneously. The excessive proliferation and migration could potentially lead to abnormal aggregation and differentiation of LECs at the lens equator, which may then accelerate the compression and hardening of the lens nucleus[13].
The manuscript was revised accordingly (Discussion section paragraph 3).
References
- Wong, T.Y.; Klein, B.E.; Klein, R.; Tomany, S.C.; Lee, K.E. Refractive errors and incident cataracts: the beaver dam eye study. Invest. Ophthalmol. Vis. Sci. 2001, 42, 1449-1454.
- Younan, C.; Mitchell, P.; Cumming, R.G.; Rochtchina, E.; Wang, J.J. Myopia and incident cataract and cataract surgery: the blue mountains eye study. Invest. Ophthalmol. Vis. Sci. 2002, 43, 3625-3632.
- Praveen, M.R.; Shah, G.D.; Vasavada, A.R.; Mehta, P.G.; Gilbert, C.; Bhagat, G. A study to explore the risk factors for the early onset of cataract in india. Eye 2010, 24, 686-694.
- Cetinkaya, S.; Acir, N.O.; Cetinkaya, Y.F.; Dadaci, Z.; Yener, H.0.; Saglam, F. Phacoemulsificatıon in eyes wıth cataract and high myopia. Arq. Bras. Oftalmol. 2015, 78, 286-289.
- Liu, C.X.; Chen, L.L. Circular rnas: characterization, cellular roles, and applications. Cell 2022, 185, 2016-2034.
- Yang, J.; Zhou, S.; Gu, J.; Guo, M.; Xia, H.; Liu, Y. Upr activation and the down-regulation of α-crystallin in human high myopia-related cataract lens epithelium. Plos One 2015, 10, e137582.
- Li, X.; Zhang, J.L.; Lei, Y.N.; Liu, X.Q.; Xue, W.; Zhang, Y.; Nan, F.; Gao, X.; Zhang, J.; Wei, J.; et al. Linking circular intronic rna degradation and function in transcription by rnase h1. Sci. China-Life Sci. 2021, 64, 1795-1809.
- Liang, S.; Dou, S.; Li, W.; Huang, Y. Profiling of circular rnas in age-related cataract reveals circznf292 as an antioxidant by sponging mir-23b-3p. Aging (Albany, NY.) 2020, 12, 17271-17287.
- Rossi, F.; Beltran, M.; Damizia, M.; Grelloni, C.; Colantoni, A.; Setti, A.; Di Timoteo, G.; Dattilo, D.; Centrón-Broco, A.; Nicoletti, C.; et al. Circular rna znf609/ckap5 mrna interaction regulates microtubule dynamics and tumorigenicity. Mol. Cell 2022, 82, 75-89.
- Du, W.W.; Yang, W.; Liu, E.; Yang, Z.; Dhaliwal, P.; Yang, B.B. Foxo3 circular rna retards cell cycle progression via forming ternary complexes with p21 and cdk2. Nucleic Acids Res. 2016, 44, 2846-2858.
- Du WW; Zhang, C.; Yang, W.; Yong, T.; Awan, F.M.; Yang, B.B. Identifying and characterizing circrna-protein interaction. Theranostics 2017, 7, 4183-4191.
- Yang, J.; Zhou, S.; Gu, J.; Guo, M.; Xia, H.; Liu, Y. Upr activation and the down-regulation of α-crystallin in human high myopia-related cataract lens epithelium. Plos One 2015, 10, e137582.
- Bassnett, S.; Costello, M.J. The cause and consequence of fiber cell compaction in the vertebrate lens. Exp. Eye Res. 2017, 156, 50-57.

Round 2
Reviewer 1 Report
The authors' responses are satisfactory. Therefore, I recommend acceptance in its present form.